# Metal Oxide-Related Dendritic Structures: Self-Assembly and Applications for Sensor, Catalysis, Energy Conversion and Beyond

**DOI:** 10.3390/nano11071686

**Published:** 2021-06-27

**Authors:** Ruohong Sui, Paul A. Charpentier, Robert A. Marriott

**Affiliations:** 1Department of Chemistry, University of Calgary, Calgary, AB T2L 2K8, Canada; 2Department of Chemical and Biochemical Engineering, Western University, London, ON N6A 5B9, Canada; pcharpen@uwo.ca

**Keywords:** dendritic metal oxide, self-assembly, sol‒gel, hydrothermal, molecular recognition

## Abstract

In the past two decades, we have learned a great deal about self-assembly of dendritic metal oxide structures, partially inspired by the nanostructures mimicking the aesthetic hierarchical structures of ferns and corals. The self-assembly process involves either anisotropic polycondensation or molecular recognition mechanisms. The major driving force for research in this field is due to the wide variety of applications in addition to the unique structures and properties of these dendritic nanostructures. Our purpose of this minireview is twofold: (1) to showcase what we have learned so far about how the self-assembly process occurs; and (2) to encourage people to use this type of material for drug delivery, renewable energy conversion and storage, biomaterials, and electronic noses.

## 1. Introduction

A dendritic structure exhibits a tree-like shape containing stems and branches. It can be a macromolecule, supramolecule, or nanostructure. Despite appearing otherwise in literature, herein “dendrites” mean the dendritic structures of crystals or metals (such as gold, silver, and copper) [1,2], while dendrimers refer to highly branched macromolecules or supramolecules.

Since the discovery of dendrimers in 1978 [3], the number of reports regarding the syntheses of these types of materials has grown exponentially. Most dendrimers reported in literature are organic oligomers, macromolecules, and supramolecules, which may contain metal- or silicon-ligand moieties [4,5,6,7,8]. Specifically, the dendritic transition-metal complexes and metallodendrimers were well discussed by Astruc and Chardac [9]. Figure 1 shows the dendrimers of polyethylene and polyglycerol, which have been graphed on ceramic membranes using ‒OSiO‒ bond as a link. The resulting composite materials were found to efficiently remove aromatic hydrocarbons and trihalogen methanes from water [10].

Not so long ago, dendritic nanostructures of metal oxides or organic‒inorganic hybrids emerged from the field of material science. These dendritic structures are composed of organized 1D or 2D inorganic particulates [11], and the latter is made of 1D or 2D macromolecules/supramolecules (Scheme 1). Noticeably, most of these materials are prepared through self-assembly via a sol‒gel or hydrothermal route, which often involves supramolecular organization via molecular recognition between the specially designed species (e.g., between the oligomers and surfactants) [12]. Chemical vapor deposition and electrodeposition are also well-established approaches for making sophisticated nanostructures [13,14], but the sol‒gel and hydrothermal methods are more scalable for commercialization. Due to their special hierarchical nanostructures, dendritic metal oxides exhibit open macropores (>50 nm) and the large mesopores (2–50 nm). This makes most of their surface area available for large molecules or causes a situation where mass transfer is the bottleneck of a process. In a water splitting process, for instance, the produced hydrogen bubbles can be trapped within the pores of electrodes, which makes it difficult for water molecules to access the active sites of the catalyst [15]. This problem can be solved by using the electrode with a dendritic nanostructure.

Dendritic metal oxides are potentially useful in the fields of medicine [16,17,18], biomaterials [19], heterogeneous catalysis [20], protective coatings [21,22,23,24], molecular machines [25], energy conversion and storage [26], sensing electronics [27,28,29], and low-voltage transistors [30,31,32,33]. Recently, another type of dendritic structure has caught people′s attention: dendritic mesopores. These radially aligned channels are generated within spherical particles [34,35]. These dendritic pores are different from the parallel mesopore channels within silicate MCM-41, a milestone work performed by Mobil Oil Cooperation [36].

The purpose of this paper is to review, from the synthesis and application perspectives, metal oxide-based dendritic nanostructures.

## 2. Synthesis of Dendritic Metal Oxides

### 2.1. Sol‒Gel Method

The first documented sol‒gel synthesis appeared in 1939, when Geffcken and Berger described their novel oxide coating technique [37]. Because sol‒gel processing is easy for scale-up and process control, this technique has been commercialized for coating thin films and making ceramic fibers and/or aerogels [38]. During the past few decades, sol‒gel technologies have made significant progress towards developing mesoporous materials, nanoparticles (0D), one-dimensional (1D), two-dimensional (2D), and three-dimensional (3D) metal oxides [39,40,41,42]. Conventional sol‒gel reactions involve hydrolysis and polycondensation of metal salts or alkoxides in a solvent medium, such as water, an organic solvent, ionic liquid, or a supercritical fluid. For making a well-defined nanostructure, one needs to control the rates of reaction between a metal precursor (such as titanium isopropoxide) and water. This can be achieved by addition of carboxylic acids, alcohols, or surfactants [43]. It was found that addition of propylene oxide in the starting materials facilitates gel formation [44,45]. The bench-top equipment for sol‒gel synthesis can be as simple as a round-bottom flask sitting on a hot plate and stirrer. Alternatively, one can use a stainless-steel view cell reactor that can be heated and pressurized to 35 megapascals [41]. By taking advantage of better mass and heat transfer in supercritical carbon dioxide (scCO_2_), self-assembly of titanium-oxo‒alkoxyl‒acetate complexes (Ti_6_O_6_(OAc)_6_(O^i^Pr)_6_) results in dendritic nanostructures of Ti_6_O_9_(OAc)_6_. The later can be calcined in air to form titania (TiO_2_) (Figure 2a,b) [46]. When isopropanol is used, interestingly, dendritic TiO_2_ can be prepared which is composed of 2D structures (Figure 2c,d) [47]. Due to a high surface area and large pore size, this type of material shows advantages over traditional catalysts for industrial applications [48,49,50]. The sol‒gel self-assembly will be discussed further in the following section.

A spin coating of sol‒gel fluid can be used for preparing dendritic zinc stannate in a thin film, which is a sensor material for detecting certain chemicals in liquified petroleum, such as ethanol, formaldehyde, and hydrogen sulfide [51]. The hierarchical structures exhibit an open surface that facilitates diffusion of target molecules in a liquid phase. In addition, the dendritic structures are more anti-fouling than microporous counterparts, even though the latter often exhibit a higher surface area. Polyethylene glycol may play multiple roles, such as in preventing precipitation of the metal precursors in sol‒gel process, as well as in avoiding shrinkage and cracking upon drying [52].

By using cetyltrimethylammonium bromide (CTAB) as a surfactant, hydrolysis and polycondensation of tetraethyl orthosilicate (TEOS) can produce a dendritic microstructure of silica with encapsulated phenolphthalein, which can form an opto-chemical sensor device [53]. Alternatively, a thin film of dendritic polymer/SiO_2_ can be prepared by polycondensation of mixed silicon alkoxide: Si(OR)_4_, R′Si(OR)_3_, and R′HSi(OR)_2_, where R and R′ stand for CH_3_ and C_2_H_5_ alkyl groups, respectively [54]. Because the R′‒Si bond is inert and hydrolysis and condensation only occur on the ‒OR ligands, anisotropic condensation can be expected from sol‒gel reactions of R′Si(OR)_3_ and R′HSi(OR)_2_. Hence, it is not surprising to observe dendritic structures from these precursors. By using tetraethyl orthosilicate and metal (Fe, Ni, and Co) chlorides as precursors, a dendritic magnetic nanocomposite structure can be obtained via a sol‒gel route, followed by annealing up to 1373 K [55]. The resulting materials can be used for medical or information storage systems.

Self-assembly also is a powerful tool for synthesizing hydrogels with an internal dendritic structure. For example, Ga^3+^ anionic metal‒organic cubes were linked together by using ammonium or amine-based ions as a binder to form dendritic supramolecules [56]. This type of material is useful for gel-chromatographic separation of dyes, and it can also be an intermediate for hybrid metal oxides.

In Table 1, both the advantages and disadvantages of sol‒gel method are summarized, which can be compared with other synthesis methods in terms of their advantage, scalability, and production cost for a commercial scale production.

### 2.2. Hydrothermal and Solvothermal Methods

The discovery of this method can be dated back to the middle of 19th century, when K.F.E. Schafthaul synthesized quartz crystals within supercritical water [57]. By taking advantage of a high temperature, high density, and high viscosity of water or other solvents above ambient pressure, one can grow both microscopic and macroscopic crystals. In the past two decades hydrothermal/solvothermal methods have been widely used for preparing metal oxides with different sizes and morphologies. The operation temperature can be diversified from as high as 673 K to as low as 353 K [58]. In the early days, Bunsen used thick-walled glass barometer tubing for his synthesis of BaCO_3_ and SrCO_3_ crystals at 473 K and 1.5 megapascal [59]. Nowadays, a Teflon-lined stainless-steel autoclave is widely available for bench-scale synthesis. The starting materials can be varied from metal alkoxide to salts and even a metal oxide itself [60].

Hydrothermal or solvothermal techniques are also useful for synthesizing dendritic TiO_2_, as reviewed by Wu and coworkers [61]. It was found that the synthesis conditions, especially the aggregation of surfactant by molecular recognition, plays an important role in controlling the size and shape of the final products. More detail regarding the nanostructure formation mechanism will be described in the following section.

Aside from TiO_2_, other metal oxides in dendritic form also have been prepared in hydrothermal autoclaves. For example, in a cosolvent of ethanol and water, Zhang et al. prepared the dendritic structures of SnO_2_ in rutile phase at 353 K by using stannous chloride as a precursor with added diammonium phosphate [62]. Based on the SEM images in Figure 3, it seemed that the product morphology was affected by the amount of phosphate. The authors claimed that (1) the pH value of the liquid phase was controlled by the amount of phosphate, which influenced the hydrolysis of stannous salt, and (2) different concentrations of phosphate could change the nucleation rates. Indeed, phosphate anions might form coordination transition metal cations [63], which would control the nucleation rate for the self-assembly process. A similar approach was used to synthesize many other dendritic oxides, such as cupric oxide [64], cuprous oxide [65], zinc oxide [66,67,68,69], cobalt oxide [70], cobalt doped nickel oxide [71], CaTiO_3_ [72], CuCo_2_O_4_ [73], and LaFeO_3_ [74].

### 2.3. Electrochemical Synthesis

Surface roughness of materials is an important factor for friction, imbibition, wetting, and surface area [75,76]. To increase surface roughness of metal and oxides, one can use an electrochemical technique [77]. For example, Jeun and coworkers have demonstrated that a dendrite form of tin oxide can be prepared via an electrical deposition route. In their experiment, they selected electrolytes containing tin sulfate and sulfuric acid. The tin foam obtained was oxidized into dendritic tin oxide after annealing in air at 973 K, as revealed under electron microscope at different magnifications (Figure 4a,b) [78].

### 2.4. Organic‒Inorganic Hybrid materials 

Nature has created remarkable organic‒inorganic composite nanostructures, such as the materials found in diatom and mollusk shells in the past hundred million years of evolution on our planet. Humans have also learned a great deal on how to synthesize composite materials and have used them in a variety of applications such as Maya blue paint, dental fillers, and the structural materials for auto and aerospace industries [79]. To achieve desired chemical and physical properties, such as plasticity (from polymers) and mechanical strength (from inorganic fillers), self-assembly techniques are utilized to synthesize well-defined organic‒inorganic composite materials [80,81]. For example, in-situ self-assembly can be carried out either by sol‒gel reactions within polymer matrix or by polymerization of organic monomers from the surface of inorganic nanoparticles. Sometimes it is possible to simultaneously synthesize both organic and inorganic components [82,83,84].

One strategy for growing polymers from metal oxides is to modify the metal oxide surface with bifunctional molecules such as methacrylic acid, which has both carboxylic group and carbon double bonds. While carboxylic group can be anchored to TiO_2_, the double bond can be used for the consequent in-situ polymerization of methyl methacrylate. As shown in Figure 5, poly(methyl methacrylate) (PMMA) can be pictured as branches on the titania domains [85]. The resulting materials provided improved dynamic Young’s moduli with potential use as dental and bone fillers. Without chemical bond between the polymers and the TiO_2_ fibers, the mechanical strength of the nanocomposite was found to be lower. This indicates that a dendritic polymer/inorganic nanocomposite has a better mechanical property than the counterpart made by a mechanical mixing of the organic and inorganic components. Besides the carboxylic functional group, phosphate groups can also be used to provide a chemical linkage between the organic and inorganic moieties [33,86,87]. Phosphorous dendrimers have been used to synthesize TiO_2_ anatase crystals at a reaction temperature as low as 333 K without further calcination [75]. This approach may open a new avenue for one-pot or in-situ synthesis of organic‒inorganic composite materials, as it avoids the normally required calcination procedure which can destroy the organic component.

### 2.5. Supramolecular Dendrimers 

Because supramolecular dendrimers with high quality and purity have potential for drug delivery, biosensor, and catalysis applications, many strategies have been explored for synthesizing these types of materials [88,89]. Here we selected an example of self-assembly of silsequioxane-based dendrimers, because the solvent′s effect on the π‒π stacking, hydrogen bonding, and van der Waals interactions were investigated [90]. In methyl methacrylate, π‒π stacking of the dendrimer, as shown in Figure 6a, resulted in formation of linear supramolecules. These linear supramolecules formed bundles and thus can be observed with electron microscopes (Figure 6b,c,f,g). These bundles twisted together into secondary spherical structures with diameters >10 μm (see SEM images in Figure 6e), and finally, these secondary structures formed a solid network through gelation (Figure 6d). The resulting materials showed improved elasticity and viscosity.

Interestingly, poly(propyleneimine) dendrimers have been used to synthesize encapsulated nanoparticles of metal and oxides [91]. By using the functional groups decorated within the dendrimers, such as amines, carboxylic acids, or hydroxyl groups, metal cations can be selectively trapped within the polymer matrix. After reduction of these cations, monodisperse nanoparticles can be obtained.

## 3. Dendritic Structure Formation Mechanism

### 3.1. Molecular Recognition for Dendritic Mesopore Formation

Micelles and reverse micelles have been widely used for making a variety of nanostructures. The driving forces for micelle formation can be from lyocratic, electrostatic or steric interactions [92]. To synthesize dendritic mesopores within silica, Yu and coworkers took advantage of a mixture of surfactants interacting with reactants: (cetyltrimethylammonium (CTA^+^) tosylate), imidazolium ionic liquid (IL), triethanolamine (TEAH_3_), triblock copolymer (F127), tetraethyl orthosilicate (TEOS), and water [93]. The aggregation of surfactant/ionic liquid micelles were believed to be responsible for the radially aligned mesopores (Figure 7). As a green solvent, the imidazolium IL behaved as a cosurfactant which reduced the critical micelle concentration of CTA^+^. It was found that the silica size was affected by the alkyl chain length of IL, but the pore size within silica was not affected. Here the triblock copolymer was used to reduce the silica particle size. The silica with dendritic mesopores have potential in drug delivery, catalysis, and peptide separation [94].

### 3.2. Sol‒Gel Anisotropic-Assembly

As mentioned earlier, dendritic nanostructures of Ti_6_O_9_(OAc)_6_ were produced by sol‒gel reactions of titanium isopropoxide with acetic acid in supercritical CO_2_ (Figure 2a,b). These structures arise due to the linear polycondensation of a reaction intermediate: Ti_6_O_6_(OAc)_6_(O^i^Pr)_6_ according to in-situ infrared and single crystal X-ray diffraction studies (Figure 8) [95,96]. The hexanuclear titanium cluster has six acetate bidentate around six titanium atoms, where the acetate groups do not undergo hydrolysis reaction. At the axial positions there are six isopropoxide that can be hydrolyzed. Condensation of these hydrolyzed complexes can lead to formation of linear oligomers and subsequent 1D nanostructures. These types of 1D structures were obtained in both supercritical CO_2_ and heptanes under similar conditions [97]; however, dendritic structures were formed only in the supercritical condition. In heptanes, agglomeration of 1D structures resulted in a 3D porous network that filled the reactor. We hypothesized that, at the initial stage of the sol‒gel process, the linear oligomers derived from Ti_6_O_6_(OAc)_6_(OH)_6_ need to have a high mobility to self-assembly into a branch structure. This is in line with the finding that both a higher diffusion and greater thermal motion of oligomers were important for controlling the self-assembly process [98,99,100]. A supercritical fluid is well-known to exhibit a low viscosity which facilitates both mass and heat transfer [101]; therefore, the dendritic structures should be more easily formed in supercritical CO_2_ than in a traditional organic solvent. The leaf-like dendritic TiO_2_ shown in Figure 2c,d was prepared in isopropanol. It was believed that the 2D structure was created through the self-assembly of a planar complex (Ti_6_O_4_(O*^i^*Pr)_8_(OAc)_8_) [47]; however, it was not understood how the dendritic shape was produced. It seems that the core area of dendritic structure (Figure 2d) has totally different morphology compared to the leaves. Indeed, we have found some unidentified crystals under transmission electron microscope. It is hoped that these crystals would shed light on the mechanism of dendritic assembly.

### 3.3. Hydrothermal Reactions

As described previously, hydrothermal and solvothermal reactions have been a popular approach for making dendritic metal oxides. Some ex-situ techniques, e.g., X-ray diffraction and electron microscopy, are helpful for nanostructure formation mechanism. In many cases, however, it is difficult to fully understand how the self-assembly process occurs. This is due to the challenge of limited in-situ techniques for a corrosive environment with elevated temperature and pressure.

Well-defined coniferous tree-like structures of iron oxide have been derived from hydrothermal reaction of potassium ferricyanide (Figure 9). In order to explain the mechanism of dendritic formation, Cao and colleagues carefully examined the orientation of the single crystals of hematite along the main stem and side branches [102]. It was found that the fast-growing main stem was along [1Ī00] direction, while the side branches were along [10Ī0] and [0Ī10] directions. According to the authors, the crystallization kinetics are related to the surface net charge in each direction of a single crystal. A net neutral facet would facilitate a faster growth; on the other hand, a net charged facet would be affect more by solvent effect, which would decrease the growth in that direction. Interestingly, the magnetic property of the dendritic α-Fe_2_O_3_ was related where it is positioned in the dendritic structure. The morin transition temperature of the main stem was found higher (243 K) than that of branches (216 K). In a separate experiment, Wen and coworkers synthesized silver dendrites. They found that the main stem and branches grew along <100> and <111> directions, respectively. In addition, it seemed that the precursor molecule diffusions also played a role in the dendrite formation [103].

During synthesis of dendritic rutile TiO_2_, Sun and coworkers studied the surfactant′s effect on the hydrothermal products. When cetyltrimethylammonium bromide (CTAB) was added to the titanium precursor with chloric acid, a dendritic structure composed of rods was produced. However, when ethylene glycol (EG) was added to the above mixture, the dendritic structure was made of nanoribbons versus rods. Furthermore, when both EG and urea were added to the mixture, the dendritic structures of nanofibers were produced (Figure 10) [104]. The authors proposed that ethylene glycol reduced the hydrolysis of the titanium precursor, and addition of urea impeded hydrolysis further, thus facilitating 2D and 1D nanostructure formation, respectively. The dendritic rutile TiO_2_ was tested as anodes for lithium-ion batteries, which showed an enhanced charge capacity and extended life span. The authors attributed the better performance to the morphological advantages of dendritic TiO_2_: (1) it facilitates the charge/discharge process of Li^+^, and (2) a fast electron transfer along the 1D/2D TiO_2_.

## 4. Applications

### 4.1. Conductometer Sensors

By using millions of olfactory receptor neurons and hundreds of functional genes, humans can distinguish thousands of odors. About 50% of people can recognize hydrogen sulfide (H_2_S), a lethal gas, at a level as low as 4.7 ppb, which is more sensitive than many current H_2_S monitors [105,106]. As shown in Figure 11, the olfactory receptor neurons contain dendrite structures, which collect odorant molecules in the back of the nasal cavity [107]. If the odorant molecules are recognized by the receptors, the latter will be activated and send an electric signal to the brain via nerve processes. Possibly inspired by the tree-like shape of olfactory receptors, many dendritic shaped biosensors have been investigated [108,109,110].

Detective sensors are essential for warning and protecting people from exposure of poisonous gases. A thin film of metal oxide semiconductors can be used as the sensing material [111], because their conductivity changes after adsorption of certain gases. For example, n-type metal oxides (CuO, SnO_2_, ZnO, and WO_3_) are often used for conductometer sensors, because their higher carrier mobility than p-type metal oxides [112,113]. However, modified p-type metal oxides may also have a high sensitivity at a room temperature [114].

As an interesting example, p-type cuprous oxide (Cu_2_O) was shown as an excellent sensing material for H_2_S, because it generates Cu_x_S on the surface after reacting with H_2_S, and Cu_x_S is known to have a high carrier mobility [115]. In addition, the detection sensitivity was significantly improved by taking advantage of accessible high surface area of nanorods and dendritic Cu_2_O, which were prepared by a sputtering process (Figure 12 and Figure 13) [116]. This result is in line with the previous findings which indicate that semiconductor morphology is an important factor to determine the detector’s sensibility [117]. The total surface electric resistance *R_T_* can be estimated as below:(1)$$\frac{1}{R_{T}}=\frac{1}{R_{S}}+\frac{1}{R_{B}}+\frac{1}{R_{BR}}$$
where *R_S_*, *R_B_*, and *R_BR_* are the resistive contributions of the surface, bulk, and dendritic structures, respectively. According to this equation, if *R_BR_* decreases significantly upon exposure to H_2_S, *R_T_* would decrease accordingly.

It is worthy to mention that fluorescent dendrimers anchored on TiO_2_ can be used as an optical sensor for detecting hazardous phenolic compounds [118].

### 4.2. Energy Conversion and Storages

Dendrimers, dendrites, and dendritic metal-oxides or sulfides are promising new materials for renewable energy, such as perovskite solar cells [119], water splitting [120], fuel cell [121], supercapacitors [122,123,124,125], as well as lithium-, sodium-, and zinc-ion batteries [126,127,128,129].

To reduce CO_2_ emissions, the governments of many countries have recently been encouraged to invest and develop hydrogen as a fuel. Electrochemical water splitting can be a solution for producing hydrogen because water is abundant; however, the most effective electrode materials involve noble metals. To reduce the amount of noble metal required for making the electrodes, one can decorate platinum quantum dots on dendritic semiconductor nanostructures, or use highly efficient dendritic nanostructures of noble metal or oxide [130].

Electrochemical water splitting involves two half reactions: hydrogen evolution and oxygen evolution. To make an electrode for oxygen evolution reaction, Oh and coworkers deposited iridium dendrites on both carbon black and the hydrothermal synthesized antimony-doped tin oxide (Figure 14) [20]. The supported dendrites showed significantly higher electrocatalytic activity than the commercial iridium black. According to the authors, the increased catalytic performance of the dendrites was due to a larger surface area and more edge and corner atoms on the dendrite surface.

### 4.3. Catalysis

Catalysts in dendritic form have been studied by many researchers [131,132,133]. After the excitement of exploring a new catalyst, it is essential to know if the dendritic form outperforms its counterparts in other morphologies, such as 0D, 1D, and 2D nanostructures. Based on specific surface area, all nanostructures are expected to have altered activity.

By comparing the effect of morphology on methanol electro-oxidation activity, Yin and coworkers found that palladium dendrites showed better performance than the corresponding nanoparticle format [134]. The authors argued that the void within the three-dimensional catalyst was the source of the activity difference between the two types of catalysts. Even though the pore size distributions were not available, the SEM images indeed showed that dendrites exhibited a larger void space (Figure 15). Larger pore size facilitates mass transfer, which is important for both reactant transport to the surface and product transport back to the bulk fluid. Hence, it is reasonable for the dendritic catalyst to outperform others. In another study, Au nanoparticles/TiO_2_ and Au dendrites/TiO_2_ were compared for their photocatalytic activity. There were no obvious specific catalytic differences between the two catalysts [135]. Nevertheless, it is noticed that the Au dendrites have a larger crystal size and thereby a smaller surface area than the particle counterpart. Therefore, Au dendrite should be more active than Au nanoparticles if the activity is on the basis of per active site or surface area of gold, i.e., if the turnover frequencies are considered equal.

### 4.4. Drug Delivery 

Dendritic metal oxides, especially modified SiO_2_ with dendritic mesopores, have been extensively studied for their medical applications such as drug delivery [136] due to their biocompatibility, low toxicity, and tunable mesostructures.

Gai and coworkers incorporated iron oxide nanoparticles to control the morphology of SiO_2_ that has dendritic pores, and used the resulting drug vehicles to deliver doxorubicin hydrochloride (DOX) into the nuclei of HeLa cells. The carboxylic groups of the drug molecules were able to form hydrogen bond with the surface OH‒Si groups on SiO_2_. Then, the drug was dissolved into phosphate buffer solution and diffused through the mesopore channels of SiO_2_. The in vitro studies showed a sustained drug release within 20 h, and the intracellular microscopic images revealed that the released DOX was active for killing tumor cells [137].

## 5. Conclusions

As an organized assembly of 1D or 2D nanostructures, dendritic metal oxide-related materials exhibit a high surface area/weight ratio and open large void space, which are important for their applications in sensing, catalysis, medicine, biomaterials, as well as energy conversion and storage devices. In designing electrode materials for lithium batteries and water splitting, for examples, attention should be paid to the morphological aspect of the semiconductive metal oxides besides the chemical compositions and crystalline phases. As described earlier, macrospores are essential for mass transfer within electrodes for water splitting process, but the mechanical strength of the materials may be reduced by introduce too many large pores. In addition, the orientation (e.g., the position of the pore openings) and the dimensions of the pore channels should make it easier for gas bubbles to escape in a liquid phase. Thus, the electrode materials must be carefully engineered. As another example, lithium‒sulfur batteries have a high expectation for electrical cars because their high electrical storage capacity; however, the electrical capacity decreased too quickly after certain rounds of battery cycling. To solve this problem, one must increase the mobility of polysulfide ions. This can be achieved by increasing electrode pore size and promoting the polysulfide solubility. Another approach is to modify the electrode surface to increase the wettability of the sulfur cathode surface with sulfur and polysulfides [138]. Future work should be carried out in these fields by using carefully designed dendritic metal oxides.

While dendritic metal oxides have been synthesized via sol‒gel or solvothermal techniques, molecular recognition is often responsible for the self-assembly process. Even though many dendritic nanostructures have already been made, their synthetic mechanism is still mainly elusive. Before we can truly design a controlled 3D structure, it is necessary to carry out more fundamental studies, including computational simulations, to provide a better understanding of both the underlying chemical reactions and self-assembly mechanisms.

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
