# Peer review of "Metal Oxide-Related Dendritic Structures: Self-Assembly and Applications for Sensor, Catalysis, Energy Conversion and Beyond"

_nanomaterials, 2021, doi:10.3390/nano11071686_

Round 1

Reviewer 1 Report

This mini-review provides a consistent overview on metal oxide dendritic nanostructures. After a survey on preparation methods, the system formation mechanism and applications are examined, with regard to the interplay between the processing conditions, material structure, nano-organization and functional performances in view of various eventual end-uses. In spite of various contribution in the fields, in my opinion this work merits publication after the authors have properly addressed the following issues:

  • The front matter of the paper (Abstract, Introduction, Conclusions) should be restructured and made more elaborated in order to present the unique characters of this review, its uniqueness with respect to the state-of-the-art in the field, and also highlight at the end the most interesting perspectives for future developments of the present research activities.
  • For completeness, the authors are suggested to include the following additional references: https://doi.org/10.1039/C2NR12083F; https://doi.org/10.1002/cphc.201701090; Nature Communications volume 9, Article number: 381 (2018); https://doi.org/10.1039/C2CE25624J; https://doi.org/10.1016/j.snb.2021.129687.
  • As a general observation, the authors are suggested to implement their figures and the related discussion in order to provide at least selective representative data on the chemical composition of the target materials and its interrelations with the pertaining functional behavior.
  • Scale-bars in Figs. 6b, 6c, 9c-g and 12 are almost unreadable.
  • Page 6, lines 179-182: please elaborate the reported observations and provide a more detailed explanation.
  • Page 9, lines 237-241: even ex-situ techniques may help. Please note.

Author Response

Thank you for your comments.

Reviewer 2 Report

A very interesting work that touches upon an important aspect of the dendritic structure. The work touches on many threads, but I miss a broader summary of the information presented. In what direction will the research be conducted further, what new applications may arise on this basis. I think the authors should complete this. Secondly, I have a few comments that I have posted below:

line 97-101, I think it is worth inserting a table with advantages and disadvantages of the sol-gel method. The same applies to the other methods for the preparation of the dendritic structure: Hydrothermal and Solvothermal Methods, Electrochemical Synthesis. This would make it easier for the reader to follow this work.

line 197-198, what exactly is the role of ionic liquid in micelle formation? Why are only ionic liquids based on the imidazolium cation used?

line 293-297, recently there have been many interesting papers on the use of metal oxide sensors in the detection of poisonous gases:

a). Emerging strategies for enhancing detection of explosives by artificial olfaction, Microchemical Journal, 164, 106025.

b). Metal oxides nanowires chemical/gas sensors: recent advances, Mater. Today Adv., 7 (2020), Article 100099.

c). Bulk detection of explosives and development of customized metal oxide semiconductor gas sensors for the identification of energetic materials, Sensors Actuators, B Chem., 258 (2018), pp. 1252-1266,

d). Miniaturised MOX based sensors for pollutant and explosive gases detection, Sensors Actuators, B Chem., 249 (2017), pp. 647-655.

line 301, what does metallic CuO mean? Please explain?

line 348-349, what was the porosity value? How did these two forms differ in terms of porosity?

Author Response

Thank you for your feedback.

Please see the attached file for my response.

Reviewer 3 Report

In this review, Sui and co-workers summarized the recent advances and remarkable studies of dendritic metal oxide-related materials, in terms of their synthetic methods, formation mechanisms and relevant applications. Metal oxide-based dendritic nanostructures have been popular topics in recent years. Thus, the manuscript of this review is timely to the field and tightly focused. Also, it is a very comprehensive information for all researchers who would like to get insight in this field.

The topic is interesting and suitable for Nanomaterials. There is one point that could be included/modified to further improve the quality of the manuscript.

(1) In the abstract, drug delivery and biomaterials are mentioned. However, there is very little information on dendritic nanostructures of metal oxides for drug delivery or biomaterial applications in the manuscript. Please revise it.

Author Response

Thank you for your comments. Please see the attached file for my response.

Reviewer 4 Report

This review manuscript is interesting and quite complete. It addresses important experimental methods and the corresponding applications. The manuscript is well written in an attractive style. It would have been extremely interesting if the authors would had also include computational and simulation work that evolved along the experimental discoveries. Nonetheless, this review work deserves publication.

Author Response

Thank you for your advice. Please see the attached file for my response.

Round 2

Reviewer 2 Report

I accept the responses of the authors of the manuscript to my comments. Secondly, significant changes were made to the manuscript. I think that after the introduced amendments, the manuscript can be recommended for further stages of evaluation.